# VMLH: Efficient Video Moment Location via Hashing

Zhifang Tan [1], Fei Dong [2], Xinfang Liu [3], Chenglong Li [1] and Xiushan Nie [1,*]

1   School of Computer Science and Technology, Shandong Jianzhu University, Jinan 250101, China
2   School of Journalism and Communication, Shandong Normal University, Jinan 250014, China
3   School of Software, Shandong University, Jinan 250101, China
*   Correspondence: niexiushan19@sdjzu.edu.cn

**Abstract:** Video-moment location by query is a hot topic in video understanding. However, most of the existing methods ignore the importance of location efficiency in practical application scenarios; video and query sentences have to be fed into the network at the same time during the retrieval, which leads to low efficiency. To address this issue, in this study, we propose an efficient video moment location via hashing (VMLH). In the proposed method, query sentences and video clips are, respectively, converted into hash codes and hash code sets, in which the semantic similarity between query sentences and video clips is preserved. The location prediction network is designed to predict the corresponding timestamp according to the similarity among hash codes, and the videos do not need to be fed into the network during the process of retrieval and location. Furthermore, different from the existing methods, which require complex interactions and fusion between video and query sentences, the proposed VMLH method only needs a simple XOR operation among codes to locate the video moment with high efficiency. This paper lays the foundation for fast video clip positioning and makes it possible to apply large-scale video clip positioning in practice. The experimental results on two public datasets demonstrate the effectiveness of the method.

**Keywords:** moment localization; video understanding; hashing; video grounding

## 1. Introduction

Given a video and a query sentence, the moment location task needs to find the start and end timestamps of the video clip that best match the query sentence. In the video-moment location task, "moment" indicates the start and end times of the query text corresponding to the video; that is, we need to obtain two time stamps of the video. Figure 1 illustrates the objective of this task through a simple example. Generally, video-moment location focuses on improving accuracy by using the fine-grained matching relationship among modes or reducing the computational cost by avoiding multiple candidate windows.

Recently, there has been much research into video-moment location. Hendricks et al. [1] proposed a moment context network, which roughly located segments by calculating the similarity between query sentences and different parts of different scales of the video. However, this method was not accurate enough and required a lot of calculation. In contrast, Gao et al. [2] proposed a model based on a sliding window, which could finetune the boundary in the window to achieve more fine-grained positioning. In order to further improve the accuracy, the cross-modal attention mechanism [3] has been widely used. For example, Xu et al. [4] had these two patterns interact at an early stage to produce semantically richer suggestions. Zhang et al. [5] proposed a network of simultaneous proposal and reasoning. In order to further reduce the computational workload related to candidate or sliding windows, Wang et al. [6] used deep reinforcement learning to automatically obtain the best window. In addition, Ghosh et al. [7] designed a method for sensing boundaries. In addition, some works in other fields have played a positive role in the video-moment location task. For example, the video object segmentation can accurately extract object information from a video [8,9]. Lu et al. [10] proposed a CO-attention

Siamese Network (COSNet), to address the unsupervised video object segmentation task from a holistic view. Object tracking tasks can assist in obtaining key object information in video [11,12]. In addition, the Modified Lazy Video Transmission Algorithm (MLVTA) proposed by Sodhro et al. [13] has laid a foundation for online video-moment location.

**Query： a man is peeling potatoes**

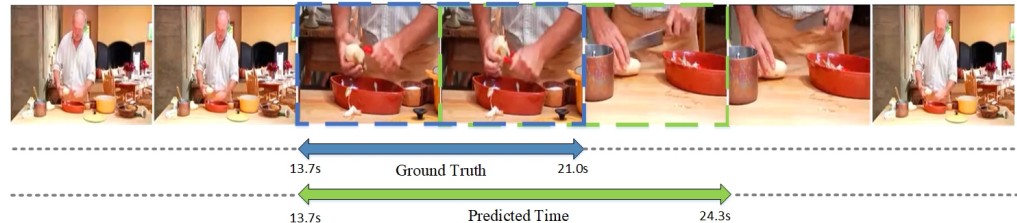

**Figure 1.** The goal of moment location is to identify a moment from a video that is most relevant to the description of the query sentence. For example, if one wants to find a clip of a man peeling potatoes, then the query can be "a man is peeling potatoes". The start and end timestamps are 13.7 s and 21.0 s, respectively, and the moment location task seeks to find the most relevant timestamps matching the query sentence.

Generally speaking, most of the existing video-moment location methods only focus on accuracy. However, with the explosive growth of multimedia data, especially video data, it becomes important to locate and retrieve large databases quickly.

In essence, video-moment location using natural language query belongs to the field of cross-modal retrieval, in which we use the query in text modality to retrieve one or more video clips. For the task of cross-modal retrieval, a common space is always needed to evaluate similarity. As is known, hashing maps data points to low-dimensional binary codes in Hamming space, which can be considered a common space for different modalities. Furthermore, the binary representation of hash codes is a widely studied solution for the fast retrieval of large-scale data. Due to the low storage requirements of binary hash codes, hashing can improve the retrieval speed and reduce memory storage, and the similarity among hash codes can be effectively calculated through fast XOR operation in Hamming space. Therefore, in this study, we use hashing to perform the moment location task.

The advantages of video-moment location using hashing are twofold. Firstly, using hash codes to store video semantic information can greatly reduce the space occupation, which has been effective for the growth of massive data in recent years. Secondly, the semantic matching between video and query sentences can be completed by bit operation of hash code, which can greatly improve the query speed.

In this study, we propose a video-moment location method with hashing (VMLH), which consists of three parts: a video-hashing network, a sentence-hashing network, and a location-prediction network. VMLH can carry out end-to-end training. In the positioning process, we only need a language query, which is flexible and can avoid the consumption of space and time.

The contributions of this study are summarized as follows:

- An efficient video-moment location method with hashing is proposed, which makes full use of hash retrieval and greatly improves the efficiency of the task.
- There is no complex interaction and fusion process in the proposed method, and videos do not need to be fed into the network during the location, which leads to higher efficiency and better scalability compared with the existing methods.

## 2. Related Work

Given that moment localization, activity localization and video retrieval with text queries in are two related tasks, we provide a brief description of them.

### 2.1. Activity Localization

Activity localization is the process of locating the start and end times of certain actions in a video. The purpose of the activity localization task is to enable the machine to recognize the actions occurring and predict when they transpire in the video automatically. There are only a few common types of actions that can be localized, such as running, jumping, throwing, etc. In contrast to video-moment location, activity location cannot be queried using natural language, and the categories of actions are limited, but its mature models are often used as backbone networks for other video tasks. Earlier work [14] localized by performing frame-level or window-level classification, followed by manual aggregation. Later, a two-stage approach [15] of proposal generation and boundary fine-tuning was used. Some models [16] now combine proposal generation and boundary fine-tuning for end-to-end training.

### 2.2. Video Retrieval with Sentence Queries

Video retrieval with text queries can obtain the entire video from a collection of videos associated with a text description. In contrast to the moment localization in this study, it does not need to predict the start and end timestamps of the moment, and its main difficulty is in learning to distinguish between different videos, rather than different parts of the same video. Currently, the dominant approach to this cross-modal retrieval is to encode different modal features into a joint embedding space to measure semantic similarity. Mithun et al. [17] encoded the video and text into global vectors. Although this global representation was efficient, it may lead to the loss of some critical details. To avoid these problems, Yu et al. [18] computed the matching relationship between words and frames and further computed the matching relationship between the entire video and the query. However, the natural language usually containing the logical structure is complex, and sometimes partial matches are not representative of the overall match relationship. For example, there is a piece of natural language with logical structure, "A man jumps from the ground for the second time", which may locate two jump clips in the video. To this end, a number of studies, such as the method in [19], have been working on this.

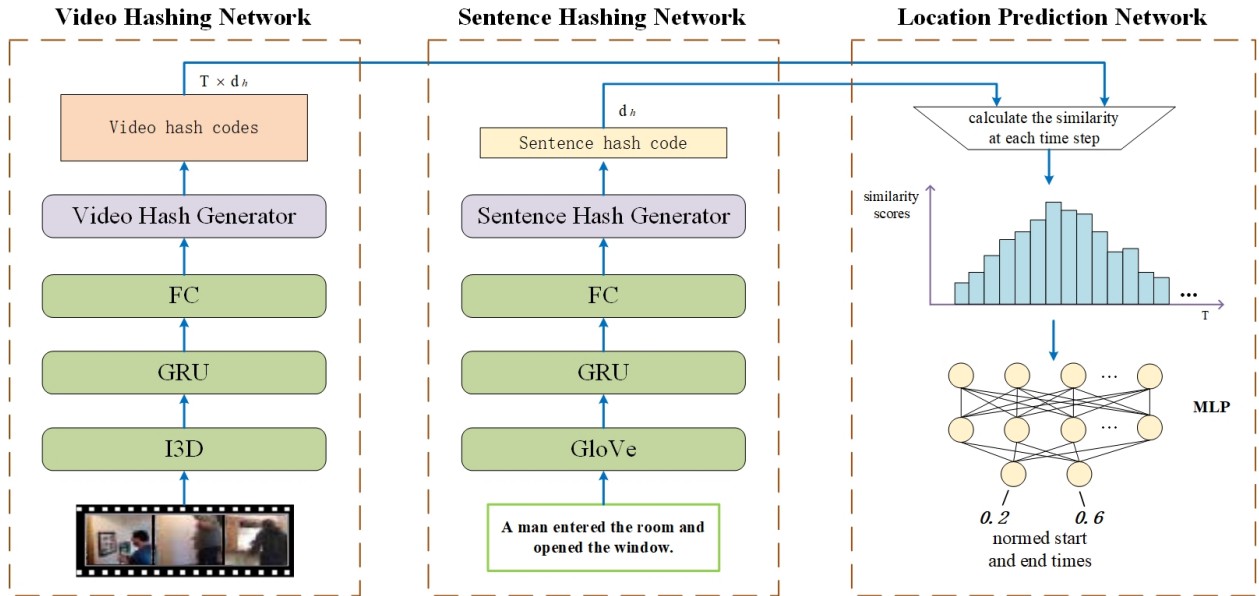

**Figure 2.** The framework of VMLH. VMLH consists of three parts: a video hashing network, a sentence hashing network, and a location prediction network.

### 3. Proposed Method

This section will first provide a description of the moment location task and then describe the proposed VMLH model and its training and reasoning process. Figure 2

shows the framework of the VMLH, which includes three components: a video-hashing network, a sentence-hashing network, and a location-prediction network. Each part will be detailed in the next section. The main framework of the model is to extract video features through the I3D network and extract text features through GloVe; then, it uses GRU to extract the advanced features of these two modes and map them to a hash matrix; next, the similarity between the two hash matrices is calculated and fed into the MLP to locate the final timestamp. The idea of the VMLH is to train a model so that it can generate a set of hash codes $\mathbf{H}_v = \{\mathbf{h}_t^v\}_{t=1}^T$ for video $v$ ($\mathbf{h}_t^v$ is the hash code of video clip $\mathbf{c}_t$, and $T$ is the total number of video clips) and generate hash code $\mathbf{h}^s$ for a query sentence $s$. Generally, the task of hashing is to convert the original video, image, or text into binary codes. The values $-1$ and $1$ are obtained with the proposed method, and we also can convert them into 0 and 1 for XOR operations. Then, using the location prediction network, the start and end timestamps are predicted by calculating the similarity among the hash codes.

### 3.1. Video Hashing Network

Given a raw video, a video encoder is used to transform it into a sequence of features $\mathbf{V} = \{\mathbf{c}_t\}_{t=1}^T$. These features are then passed through a bidirectional gated recurrent unit (GRU) [20] network to mine the timing information, and a fully connected layer (FC) with activation functions is used to generate a real vector at each moment. Finally, the hash code $\mathbf{h}_t^v$ is obtained as follows:

$$\mathbf{h}_t^{g1} = GRU(\mathbf{c}_t, \mathbf{h}_{t-1}^{g1}), \tag{1}$$

$$\mathbf{r}_t^v = Tanh(\mathbf{W}_\alpha \mathbf{h}_t^{g1} + \mathbf{b}_\alpha), \tag{2}$$

$$\mathbf{h}_t^v = sign(\mathbf{r}_t^v), \tag{3}$$

where $\mathbf{h}_t^{g1}$ is the concatenation from the bidirectional output of the GRU at time step $t$, and $\mathbf{r}_t^v$ refers to an undiversified vector of real numbers. The symbols $\mathbf{W}_\alpha$ and $\mathbf{b}_\alpha$ are the learnable matrix and bias, respectively.

### 3.2. Sentence-Hashing Network

The structure of the sentence hashing network is almost identical to that of the video hashing network, except that only the output of the last time step of the GRU is used as the overall semantics of the sentence. A query sentence with $N$ words can be represented as $\mathbf{S} = \{\mathbf{w}_n\}_{n=1}^N$, after extracting the features using GloVe [21], and the hash code $\mathbf{h}^s$ of the query sentence is obtained as follows:

$$\mathbf{h}_n^{g2} = GRU(\mathbf{w}_n, \mathbf{h}_{t-1}^{g2}), \tag{4}$$

$$\mathbf{r}^s = Tanh(\mathbf{W}_\beta \mathbf{h}_N^{g2} + \mathbf{b}_\beta), \tag{5}$$

$$\mathbf{h}^s = sign(\mathbf{r}^s), \tag{6}$$

where $\mathbf{h}_n^{g2}$ is the concatenation from the bidirectional output of the GRU at time step $n$, and $\mathbf{r}^s$ represents the real vectors of the output of the last time step $N$. The symbols $\mathbf{W}_\beta$ and $\mathbf{b}_\beta$ are the learnable matrix and bias, respectively.

### 3.3. Location Prediction Network

The role of the location prediction network is to calculate the corresponding start and end times of the moment based on the distribution of the similarity scores. Specifically, the similarity score $\mathbf{s}_t^h$ at each time step is calculated by the following formula:

$$\mathbf{s}_t^h = Sigmoid(\mu \mathbf{h}_t^v \cdot \mathbf{h}_s), \tag{7}$$

where $Sigmoid(\cdot)$ is a sigmoid function. The symbol $\mu$ is the deflation factor to prevent the similarity values from straying too far from the origin and causing the gradient to

disappear. Subsequently, the similarity scores $\mathbf{s}_t^h$ for each time step are collapsed into a vector $\mathbf{s}^h$ and fed into a multilayer perceptron. Then, using the multilayer perceptron, we obtain a start and end timestamp as follows:

$$\mathbf{h}^f = Tanh(\mathbf{W}_\gamma \mathbf{s}^h + \mathbf{b}_\gamma), \tag{8}$$

$$\mathbf{l} = \mathbf{W}_\zeta \mathbf{h}^f + \mathbf{b}_\zeta, \tag{9}$$

where $\mathbf{W}_\gamma$, $\mathbf{W}_\zeta$, $\mathbf{b}_\gamma$, and $\mathbf{b}_\zeta$ are learnable parameters. The vector $\mathbf{l}$ represents the predicted normative moment, and it consists of two items $l^s$ and $l^e$, which represent the predicted start and end times, respectively.

### 3.4. Training and Inference

Given a video and a query sentence whose corresponding start and end times are $t_s$ and $t_e$, respectively, the ground truth score $s_t^g$ at time $t$ for this duration corresponds to 1, and the remainder corresponds to 0. $s_t^r$ is the similarity score of time $t$ obtained by our method. We move the semantically similar fragments of sentence and video closer together in Hamming space. The similarity loss $L_s$ is:

$$L_s = -\frac{1}{T} \sum_{t=1}^{T} s_t^g log(s_t^r) + (1 - s_t^g)log(1 - s_t^r). \tag{10}$$

A smoothed L1 loss function $R(\cdot)$ [22] is used to calculate the location loss $L_l$:

$$L_l = R(l^s - t_s^n) + R(l^e - t_e^n), \tag{11}$$

where $t_s^n$ and $t_e^n$ are the start and end times after normalization, respectively. Finally, the entire loss function is described as follows:

$$L = L_s + \lambda L_l, \tag{12}$$

where $\lambda$ is a parameter.

The inference process can be completed end-to-end, or it can first generate the hash code for storage through the corresponding network and then use the location-prediction network for matching when necessary.

After obtaining $l^s$ and $l^e$, the predicted start and end timestamps $t_s^*$ and $t_e^*$ can be obtained by calculating the following:

$$t_s^* = l^s \times duration, \tag{13}$$

$$t_e^* = l^e \times duration, \tag{14}$$

where *duration* indicates the time duration of the entire video.

To better represent the complete training process, the algorithm for training the VMLH is shown in Algorithm 1. The overall flow chart of this method is shown in Figure 3.

---

**Algorithm 1:** Learning algorithm for VMLH

---

**Input:** Training video $\mathbf{V} = \{\mathbf{c}_t\}_{t=1}^{T}$ and the query sentence $\mathbf{S} = \{\mathbf{w}_n\}_{n=1}^{N}$. The ground truth timestamps $t_s^n$ and $t_e^n$.

**Output:** Predicted video start and end timestamps $t_s^*$ and $t_e^*$.

**Initialization:** Initialize the model with the pretrained I3D and GloVe parameter file.

**Repeat:** Randomly sample a minibatch of video from $\mathbf{V}$, obtain the corresponding sentence $\mathbf{S}$ according to the video ID, and perform the following operations:

1 Obtain the video feature matrix $\mathbf{c}_t$ and the corresponding text feature matrix $\mathbf{w}_n$.

2 For each clip feature $\mathbf{c}_t$, obtain the video hash codes $\mathbf{h}_t^v$. Obtain the hash codes $\mathbf{h}^s$ of sentence feature $\mathbf{w}_n$.

3 XOR the hash codes of each video clip $\mathbf{h}_t^v$ with the sentence hash codes $\mathbf{h}^s$ to obtain the similarity score $\mathbf{s}_t^h$. Feed the similarity score into the multilayer perceptron to obtain the start and end time stamps.

4 Calculate $L_s$ and $L_l$, according to the similarity score $\mathbf{s}_t^h$ and $t_s^n$ and $t_e^n$ in the INPUT.

5 Update the parameters by utilizing backpropagation.

**Until:** a fixed number of iterations

---

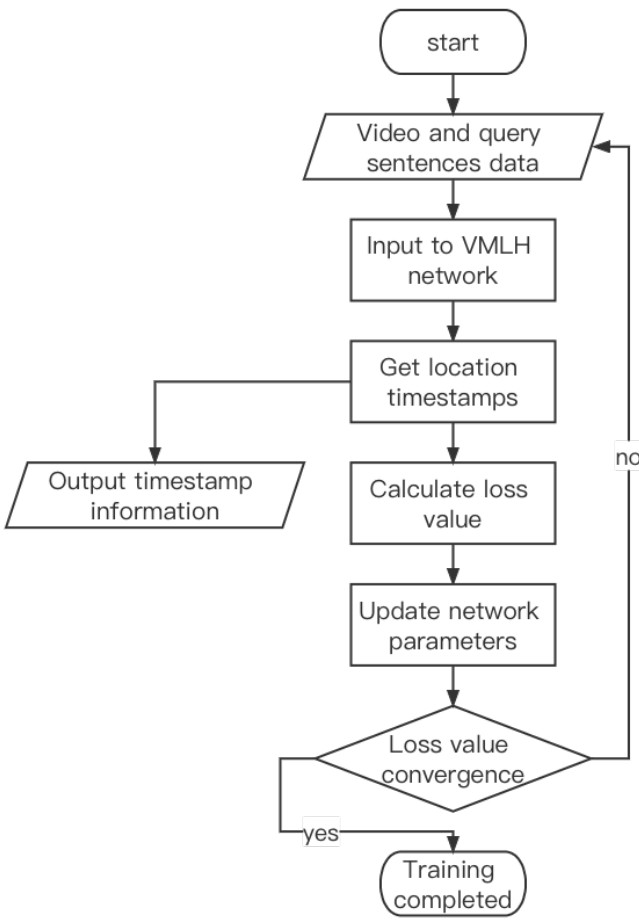

**Figure 3.** VMLH Training Flow Chart.

## 4. Experiments

### 4.1. Datasets

We evaluated the proposed VMLH on two widely used benchmark video datasets, Charades-STA [2] and ActivityNet Captions [23].

### 4.1.1. Charades-STA

The Charades-STA dataset [2] is annotated by the semi-automatic method. The average length of sentences is 8.6 words, and there is no complex logical structure. As a result, the sentences are simpler in form and usually not very long. This dataset includes 9848 tagged videos, each lasting approximately 30 s, showing the behavior of 267 different people on three continents.

### 4.1.2. ActivityNet Captions

In this dataset, each sentence corresponds to a moment of the video, which can be anywhere from a few seconds to over a hundred seconds in length. The sentences themselves are also highly complex, can be very long, and can contain multiple consecutive actions. On average, each of the 20 k videos in an ActivityNet caption contains 3.65 sentences.

### 4.2. Experimental Settings

Except as specifically mentioned, the hyperparameter settings were identical for both datasets. All the experiments of the model efficiency were run in an Nvidia TITAN Xp GPU on Ubuntu 16.04 with 256 GB memory. The video and sentence hash codes were both 64-bit. The 500-dimensional C3D [24] features and 1024-dimensional I3D [25] features were used in the datasets ActivityNet and Charades-STA, respectively. The output dimension of the sentence encoder GloVe was 300. The hidden layer sizes of the LSTMs and FCs were 256 and 128, respectively, regardless of whether they were in a video or a sentence hash network. In addition, $\lambda$ and $\mu$ were set to 0.01 and $1/6$, respectively. Moreover, we sampled the videos on the Charades-STA and ActivityNet evenly to 64 clips and 128 clips, respectively. All experiments were conducted using the Adam optimizer with a learning rate of 0.001 and a batch size of 64 for 50 epochs in PyTorch. In building the project code, we used the framework of Pytorch and Torch Lightning, and we used Wisdom to visualize the training process.

### 4.3. Evaluation Metrics

The IoU metric denotes the intersection of the predicted and ground truth moment over their union. For a fair comparison, we adopted "R@$n$, IoU = $m$" as the evaluation metric for our study. Specifically, "R@$n$, IoU = $m$" is defined as the percentage of queries having at least one result satisfying IoU $\leq m$ in the top $n$ results. Specifically, R@1 means that only one video segment was predicted. Note that our VMLH provided only one pair of timestamps, so that $n = 1$ in the experiment we reported.

### 4.4. Accuracy Performance

Tables 1 and 2 show the comparison between the VMLH and other state-of-the-art methods. The experimental results using real features were additionally provided for both datasets. The experimental results demonstrated that our proposed model had higher accuracy compared to other state-of-the-art methods even in the absence of earlier feature interaction. Moreover, using discrete hash codes in training, the location prediction network avoided inconsistencies between the training and test data types without significant loss of accuracy.

**Table 1.** R@1 performance comparison for the ActivityNet Captions dataset (%).

| Comparison | Cross–Merge Ratio | | |
|:---:|:---:|:---:|:---:|
| **Method** | **IoU = 0.3** | **IoU = 0.5** | **IoU = 0.7** |
| MCN [1] | 39.35 | 21.36 | 6.43 |
| CTRL [26] | 47.43 | 29.01 | 10.34 |
| TGN [3] | 45.51 | 28.47 | - |
| TripNet [27] | 48.42 | 32.19 | 13.93 |
| ACRN [28] | 49.70 | 31.67 | 11.25 |
| VMLH | **52.15** | **34.50** | **17.16** |

**Table 2.** R@1 performance comparison for the Charades-STA dataset (%).

| Comparison | Cross–Merge Ratio | |
|:---:|:---:|:---:|
| **Method** | **IoU = 0.5** | **IoU = 0.7** |
| CTRL [2] | 23.63 | 8.89 |
| MLVI [4] | 35.60 | 15.80 |
| ACL-K [29] | 30.48 | 12.20 |
| ACRN [28] | 20.26 | 7.64 |
| SM-RL [6] | 24.36 | 11.17 |
| QSPN [4] | 35.60 | 15.80 |
| TripNet [27] | 36.61 | 14.50 |
| VMLH | **43.80** | **20.32** |

*4.5. Model Efficiency*

Table 3 shows the efficiency comparison between our model and the other models in a single run time. The single run time is the average time taken to locate one moment in a video. VMLH-full means that neither the sentence nor the video was pre-stored as hash codes. VMLH-vh represents the video was pre-stored as hash codes, while the sentences had to move through the sentence hashing network. VMLH-h means both used hash for retrieval. Given that using hash for retrieval only requires going through the location prediction network, the model's computation was reduced by a factor of 10. When large batches were considered, the average time per retrieval required was reduced to the level of microseconds.

**Table 3.** Comparison of the models' efficiency for a single run time (s).

| Method | Single Run Time (s) |
|:---:|:---:|
| CTRL [2] | 3.41 |
| ACRN [28] | 4.42 |
| ABLR [30] | 0.06 |
| CMHN [31] | 0.0076 |
| VMLH-full | 0.0093 |
| VMLH-vh | 0.0036 |
| VMLH-h | **0.0007** |

*4.6. Ablation Experiment*

We performed ablation experiments on two datasets. Figure 4 shows the results of the location efficiency using hash codes of different lengths and without hash generators. We also used a single run time to evaluate the positioning efficiency. According to the experimental results, the hash generator greatly improved the localization efficiency. We conducted experiments on the influence of the hash code length on the precision, and the experimental results are shown in Table 4. The length of the hash code affected the representation ability of the features. We conducted precision experiments on the settings of 32-bit, 64-bit, and 128-bit hash codes. We chose the 64-bit hash code. Although the accuracy was slightly lower, the location efficiency was greatly improved.

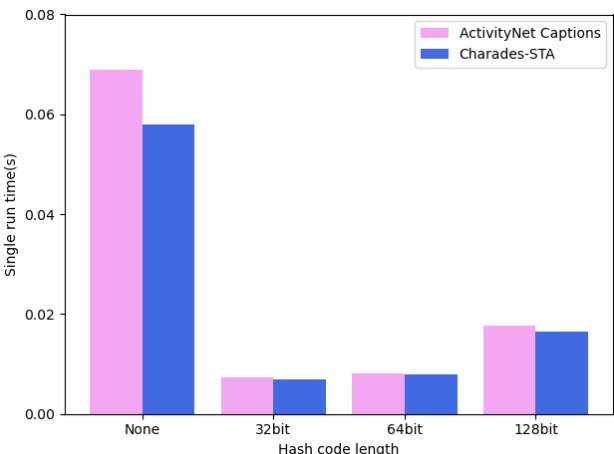

**Figure 4.** Effects of different lengths of hash codes and of not using hash generators on the location efficiency.

**Table 4.** Hash code length ablation experiment.

| Hash Code | Charades-STA | | ActivityNet Captions | | |
|---|---|---|---|---|---|
| Length | IoU = 0.5 | IoU = 0.7 | IoU = 0.3 | IoU = 0.5 | IoU = 0.7 |
| 32 bit | 42.29 | 20.16 | 51.00 | 33.74 | 16.63 |
| 64 bit | 43.80 | 20.32 | 52.15 | 34.50 | 17.16 |
| 128 bit | 43.99 | 20.76 | 52.87 | 34.36 | 17.20 |

### 4.7. Convergence Analysis

Experiments were performed on the two datasets to evaluate the convergence performance of the proposed VMLH. In the experiments described in this section, we used the relative losses to evaluate the convergence of the VMLH. Figure 5 shows that as the iterations increased, the relative loss became fairly small and stable. The convergence experiments showed that the VMLH reached convergence quickly during training, which greatly reduced the training time.

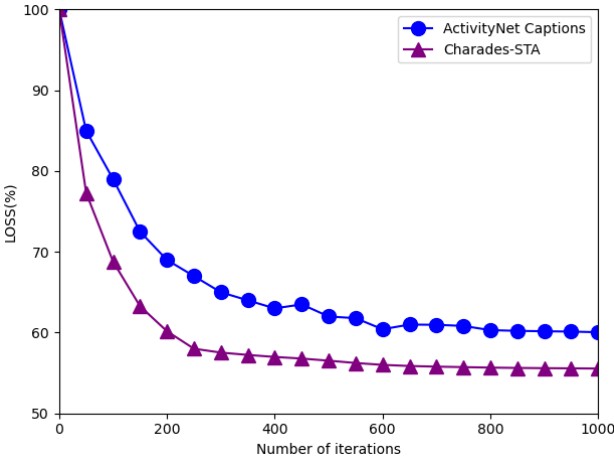

**Figure 5.** Convergence curve. Relative loss results of the VMLH on the two datasets after 1000 iterations.

### 4.8. Qualitative Results

Figure 6 shows three visual predictions on the Charades-STA dataset. The area represented by the blue line is the ground truth; the green line is the predicted segment. We

expect to estimate the start and end timestamps closer to the ground truth. The three examples in Figure 6 show that the prediction result has a high IoU on the Charades-STA dataset. However, in the first example, the start and end timestamps of the prediction have errors compared to the ground truth. Therefore, the first example results are unsatisfactory and may lead to considerable errors in the prediction results on long videos.

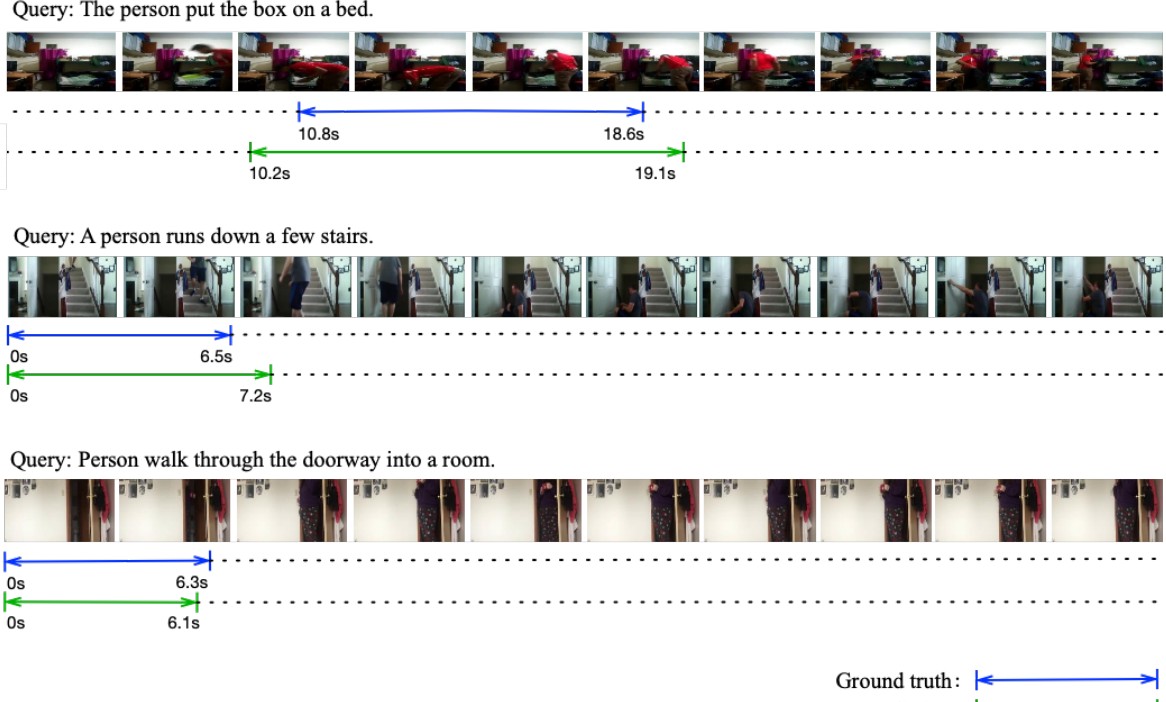

**Figure 6.** We randomly selected three location results for visual output on the Charades-STA dataset. In addition, there is a high IoU result on this dataset.

## 5. Conclusions

In this study, we proposed hashing to solve the moment location problem. The proposed VMLH model is efficient, reducing the storage space with considerable accuracy. Our proposed method still contains limitations: 1. storing hash codes in advance video and sentence hashing networks may cause additional memory consumption. 2. The XOR operation using hash codes may fail to fuse the modes, which will cause the loss of semantic information between the two models. Therefore, further improving the accuracy without early feature interaction requires more research. In addition, considering the short video duration used for training in these two datasets, the performance with long videos requires investigation in future studies. Through the ablation experiment, we concluded that the hash code as the video feature affects the accuracy. In future works, we will conduct in-depth research on this; we will enable the hash codes to learn better feature representation. This aspect will immensely improve the accuracy of our method.

**Author Contributions:** Conceptualization, Z.T.; Data curation, X.L.; Formal analysis, X.N.; Funding acquisition, X.N.; Investigation, F.D.; Methodology, X.L.; Project administration, X.N.; Resources, C.L.; Software, F.D. and C.L.; Supervision, X.N.; Validation, X.N.; Writing—original draft, Z.T.; Writing—review and editing, X.L. and X.N. All authors have read and agreed to the published version of the manuscript.

**Funding:** This work was supported in part by the National Natural Science Foundation of China (62176141, 62102235), the Shandong Provincial Natural Science Foundation for Distinguished Young Scholars (ZR2021JQ26), the Shandong Provincial Natural Science Foundation (ZR2020QF029), and the Taishan Scholar Project of Shandong Province (tsqn202103088).

**Data Availability Statement:** Not applicable.

**Conflicts of Interest:** The authors declare no conflicts of interest.

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
