# Peer review of "VMLH: Efficient Video Moment Location via Hashing"

_electronics, doi:10.3390/electronics12020420_

Round 1

Reviewer 1 Report

Authors have highlighted the emerging and core issue, but still there are various issues to be fixed.

Minor Reviews

·       Spell out each acronym the first time used in the body of the paper. Spell out acronyms in the Abstract by extending it.

·       The abstract can be rewritten to be more meaningful. The authors should add more details about their final results in the abstract. Abstract should clarify what is exactly proposed (the technical contribution) and how the proposed approach is validated.

Major Revies

·       What is the motivation of the proposed work?

·       Introduction needs to explain the main contributions of the work clearer.

·       The novelty of this paper is not clear. The difference between present work and previous Works should be highlighted.

·       Authors must explain in detail the introduction section.

·       Authors must develop the framework/architecture of the proposed methods

·       There is need of flowchart and pseudocode of the proposed techniques

·       Proposed methods should be compared with the state-of-the-art existing techniques

·       Research gaps, objectives of the proposed work should be clearly justified.

To improve the Related Work and Introduction sections authors are highly recommended to consider these high-quality research works < Medical Quality of Service Optimization over Internet of Multimedia Things >

·       English must be revised throughout the manuscript.

·       Limitations and Highlights of the proposed methods must be addressed properly

·       Experimental results are not convincing, so authors must give more results to justify their proposal.

Finally, paper  needs major changes

Author Response

TRANSLATE with x English

Arabic Hebrew Polish
Bulgarian Hindi Portuguese
Catalan Hmong Daw Romanian
Chinese Simplified Hungarian Russian
Chinese Traditional Indonesian Slovak
Czech Italian Slovenian
Danish Japanese Spanish
Dutch Klingon Swedish
English Korean Thai
Estonian Latvian Turkish
Finnish Lithuanian Ukrainian
French Malay Urdu
German Maltese Vietnamese
Greek Norwegian Welsh
Haitian Creole Persian  

TRANSLATE with COPY THE URL BELOW Back EMBED THE SNIPPET BELOW IN YOUR SITE Enable collaborative features and customize widget: Bing Webmaster Portal Back

TRANSLATE with x English

Arabic Hebrew Polish
Bulgarian Hindi Portuguese
Catalan Hmong Daw Romanian
Chinese Simplified Hungarian Russian
Chinese Traditional Indonesian Slovak
Czech Italian Slovenian
Danish Japanese Spanish
Dutch Klingon Swedish
English Korean Thai
Estonian Latvian Turkish
Finnish Lithuanian Ukrainian
French Malay Urdu
German Maltese Vietnamese
Greek Norwegian Welsh
Haitian Creole Persian  

TRANSLATE with COPY THE URL BELOW Back EMBED THE SNIPPET BELOW IN YOUR SITE Enable collaborative features and customize widget: Bing Webmaster Portal Back

Reviewer 2 Report

This paper reports on an approach to video segment location based on a natural language description based on data-dependent hashing neural networks. In terms of topic, the manuscript is suited to the Journal. In terms of methodological approach and presentation, the authors are suggested to consider the possibility to address the following remarks:

1. The term “video moment location”, which is not widely established in the field, may appear ambiguous. Since the noun “moment” typically implies a short time interval, it would be useful for the reader if the authors explained why they decided to use this noun, instead of e.g., “segment” (as in “video segment location”).

2. Section 1.1 which compares the tasks of video moment location and activity location needs the following clarifications:

2.1. Why “in contrast to video moment location, activity location cannot be queried using natural language” (p. 2, l. 64-65)?

2.2. Although it is clear that “the categories of actions are limited” (p. 2, l. 65), the authors should explain how this property of limited categories is not also conceptually present in the video moment location task.

3. Please clarify what is the “logical structure” (p. 3, l. 82) of natural language. To the reader, this phrase may appear as unjustifiably combining two fundamentally different notions of semantic proposition and syntactical structure. Thus, the authors should be more terminologically specific on this.

4. The authors explained the novelty of their approach with respect to other approaches to video moment location. It would be useful if they also emphasized the novelty of their approach with respect to other approaches engaging deep hashing neural networks for similarity search.

5. Datasets:

5.1. In Section 3.1.1, the authors state that sentences in the Charades-STA dataset are “simpler in form and usually not very long” (p. 5, l. 125-126) . In Section 3.1.2, they state that the sentences in the ActivityNet Captions dataset  are “highly complex, and can be very long and contain multiple consecutive actions” (p. 5, l. 131-132) . The notions of “simple in form” and “highly complex” sentences should be either clarified or excluded. In addition, the above statements should be quantified (e.g., the mean numbers and standard deviations of words and actions per sentence).

5.2. The authors state that “on average, each of the 20k videos in an ActivityNet caption contains 3.65 sentences” (p. 5, l. 133). The standard deviation should be provided.

6. How the values of the deflation factor (i.e., 1/6, cf. Eq. (7)) and the parameter lambda (0.01, cf. the loss function, Eq. (12)) were determined?

7. Applying IoU metrics, i.e., the Jaccard distance, to conceptualize the accuracy of the reported approach represents a popular choice. To perform a “fair comparison” of the proposed approach to other state-of-the-art approaches, the authors consider “the percentage of queries having at least one result satisfying IoU ≤ m in top n results”, where n is set to 1 (p 6., l. 148-150) and show that their approach has higher accuracy. However, since any given IoU score may correspond to many different video segmentations, the reported results are hard to interpret with respect to the actual “standard” accuracy of the system. Thus, it would be useful if the authors provided the distribution (e.g., at least, the mean and standard deviation) of the IoU scores obtained when the proposed approach was applied on the considered datasets.

Author Response

(The authors gave the same response as above.)

Reviewer 3 Report

Comment:

In this paper, the authors proposed VMLH, a video processing framework, for moment location. Their framework is based on hashing algorithm and GRU network, which can predict the corresponding timestamp of a target video frame according to the similarity. Based on their experimental validation, the VMLH framework achieved better accuracy and lower time consumption compared to the conventional methods. For my personal optional, I think the works in the paper are interesting and meaningful, but the experimental validation & technical presentation are not comprehensive and accurate enough. Therefore, I would suggest they make an extensive revision to their manuscript before formally publishing.

To sum up,

-Advantages:

1. The time consumption performance is competitive.

2. The methodology presentation is clear.

3. The method in the paper may have good application prospects.

-Major disadvantages:

4. Will the length of the hash code influence the accuracy performance? The author did an ablation study for time consumption. I would like to know if the length will also impact the accuracy. Please supply some experiments.

5. I would hope to see the GPU memory computation of the different methods. The speed performance is good, but will the current method cost more memory than the conventional method? Please give a comparison.

6. What is the R@1 metric used for evaluating the accuracy performance? Although the author gave an explanation about it, I think most of readers (include me) are still quite confuse. Please introduce it more and include some reference.

-Small flaws:

7. Is there any limitation in the current method? Will future work or possible improvement can improve these limitations?

8. I suggest the authors show some images of their obtained results. That would help the reader to gain a more intuitive understanding.

9. There are some small typesetting mistakes. Eg. For Fig 2, there sould be a “:” in “consists of three parts: …”. Please check the manuscript carefully.

Author Response

(The authors gave the same response as above.)

Round 2

Reviewer 1 Report

Authors have improved the article, but still major sections are weak

1. Results section is weak, so more graphs can be added 

2. Related Work must be created separately as 'Section 2'....by addressing the limitations and gaps of existing works and novelty of the conducted research

3. Limitations of the proposed work must be added in the 'Conclusion' section

Author Response

TRANSLATE with x English

Arabic Hebrew Polish
Bulgarian Hindi Portuguese
Catalan Hmong Daw Romanian
Chinese Simplified Hungarian Russian
Chinese Traditional Indonesian Slovak
Czech Italian Slovenian
Danish Japanese Spanish
Dutch Klingon Swedish
English Korean Thai
Estonian Latvian Turkish
Finnish Lithuanian Ukrainian
French Malay Urdu
German Maltese Vietnamese
Greek Norwegian Welsh
Haitian Creole Persian  

TRANSLATE with COPY THE URL BELOW Back EMBED THE SNIPPET BELOW IN YOUR SITE Enable collaborative features and customize widget: Bing Webmaster Portal Back

Reviewer 2 Report

In the revised manuscript, the authors have addressed the remarks from my previous review report.  Although the remarks related to the activity location task and the corpora description are addressed rather formally, the manuscript has been improved.

As a suggestion for further improvement, I propose to the authors to consider the possibility to address the following remark:

1. The response to remark 2.1 from the previous review report is not adequate. The authors should either substantially clarify their statement that “activity location cannot be queried using natural language” (p. 3, l. 77) or to remove the statement form the manuscript.

Author Response

(The authors gave the same response as above.)

Reviewer 3 Report

The revised manuscript looks better. However, I still suggest the authors supply a discussion about the limitation of the current method. Future work or possible improvement should also be addressed. 

Author Response

(The authors gave the same response as above.)

Round 3

Reviewer 1 Report

Paper is improved, so minor changes are required 

Reviews for Authors

1. Limitations of proposed method must be rewritten 

2. Paper must be proof-read by native English speaker 

Author Response

(The authors gave the same response as above.)
